# An integrated in vivo/in vitro framework to enhance cell-free biosynthesis with metabolically rewired yeast extracts

Blake J. Rasor [1,2,3], Xiunan Yi[4], Hunter Brown[1,2,3], Hal S. Alper [4,5✉] & Michael C. Jewett [1,2,3,6,7✉]

Cell-free systems using crude cell extracts present appealing opportunities for designing biosynthetic pathways and enabling sustainable chemical synthesis. However, the lack of tools to effectively manipulate the underlying host metabolism in vitro limits the potential of these systems. Here, we create an integrated framework to address this gap that leverages cell extracts from host strains genetically rewired by multiplexed CRISPR-dCas9 modulation and other metabolic engineering techniques. As a model, we explore conversion of glucose to 2,3-butanediol in extracts from flux-enhanced *Saccharomyces cerevisiae* strains. We show that cellular flux rewiring in several strains of *S. cerevisiae* combined with systematic optimization of the cell-free reaction environment significantly increases 2,3-butanediol titers and volumetric productivities, reaching productivities greater than 0.9 g/L-h. We then show the generalizability of the framework by improving cell-free itaconic acid and glycerol biosynthesis. Our coupled in vivo/in vitro metabolic engineering approach opens opportunities for synthetic biology prototyping efforts and cell-free biomanufacturing.

[1] Department of Chemical and Biological Engineering, Northwestern University, Evanston, IL, USA. [2] Chemistry of Life Processes Institute, Northwestern University, Evanston, IL, USA. [3] Center for Synthetic Biology, Northwestern University, Evanston, IL, USA. [4] Institute for Cellular and Molecular Biology, The University of Texas at Austin, Austin, TX, USA. [5] McKetta Department of Chemical Engineering, The University of Texas at Austin, Austin, TX, USA. [6] Robert H. Lurie Comprehensive Cancer Center, Northwestern University, Chicago, IL, USA. [7] Simpson Querrey Institute, Northwestern University, Chicago, IL, USA. ✉email: halper@che.utexas.edu; m-jewett@northwestern.edu

Microbial bioconversion forms the cornerstone of the biotechnology industry with applications spanning from the food and beverage sector[1] to the chemical, fuel, and nutraceutical industries[2,3]. Among the choice of microbes, the yeast *Saccharomyces cerevisiae* serves an important role as both a model host organism[4–7] and a suitable industrial biocatalyst[8]. Through metabolic engineering, this host can be tailored for in vivo production of numerous heterologous products[8], including the biofuel butanol[9], antioxidant nutraceutical resveratrol[10], versatile monomer 3-hydroxypropionic acid[11], anti-malarial drug precursor artemisinin[12], and diverse bioactive alkaloids[13–16], among others. These production strains are substantially modified through both heterologous pathway complementation and rewiring of metabolic pathways to increase pools of precursor compounds, downregulate or eliminate competing pathways, and/or optimize cofactor regeneration routes[2,3,17]. Despite the success of these approaches, in vivo biosynthesis of chemical products directly competes with cell growth for carbon and energy while withstanding the constant metabolic burden of producing nonessential metabolites. This creates selective pressure against heterologous or overexpressed pathways[18]. Balancing this pressure to push forward innovative in vivo biochemical production platforms is a time- and labor-intensive challenge.

In contrast to in vivo biosynthesis, many seminal studies of fundamental biological phenomena have embraced cell-free approaches to elucidate mechanisms behind the genetic code[19], eukaryotic translation[20,21], and fermentation by yeast[22,23], devoid of growth constraints. Platforms for cell-free gene expression (CFE)[24–26] and cell-free metabolic engineering (CFME)[27–31] have recently matured as complementary approaches for bio-discovery and the design of both enzymes and metabolic pathways. Without the cellular impediments of homeostatic maintenance, cell division, and membrane transport, in vitro metabolic systems enable the rapid assembly and testing of large combinations of biosynthetic enzymes. This ultimately provides a biological analogue of combinatorial synthetic chemistry with similar implications for scalability and throughput[32–34]. Such systems have been applied to a growing repertoire of metabolic pathways, including 2,3-butanediol (BDO)[35,36], butanol and isobutanol[33,37], 3-hydroxybutyrate[38], styrene[39], the monoterpenes limonene, bisabolene, and pinene[34,40], and cannabinoids[41], among others[28,32]. Cell-free systems also offer the ability to explore biosynthetic capabilities using the maximum catalytic rate of enzymes[27,29] as well as toxic molecules and chemical conditions outside homeostatic ranges[42], such as untreated substrates[35], cytotoxic products[43], and cytotoxic concentrations of product[39]. In the case of *S. cerevisiae*, extracts have been evaluated for the facile assessment of enzyme activity in fatty acids biosynthesis[44] and propanediol production[45] as well as bio-ethanol production at elevated temperatures[46], but the biosynthetic potential of yeast extract remains mostly unexplored for value-added chemical products.

We posit that combining cell-based metabolic engineering for strain optimization with in vitro metabolism platforms can accelerate the ability to build, study, and optimize biosynthetic pathways. Specifically, in vivo systems offer genetic tractability to manipulate and optimize cellular metabolism, while cell-free systems offer a high degree of flexibility to vary enzyme stoichiometry, study enzyme variants, alter cofactor conditions, and assess substrate specificity[27,34,47,48].

Here, we set out to create an integrated cellular/cell-free metabolic engineering workflow coupling in vivo genetic rewiring with in vitro activated metabolism. We first established a metabolically active yeast extract capable of converting glucose to both native and heterologous metabolites (ethanol and BDO, respectively). Then

using CRISPR-dCas9 rewiring, we demonstrated that extracts constituted from altered cells can exhibit productivities exceeding 10 mM/h and obtain titers of nearly 100 mM BDO (0.9 g/L-h and 9 g/L, respectively). This titer is nearly 3-fold greater than the unmodified extract, and the productivity is comparable to that observed in living cells when normalized on the basis of cell mass. Finally, we exhibited the flexibility and effectiveness of this integrated metabolic engineering approach by showing improved productivities for two additional metabolites, itaconic acid, and glycerol, by generating extracts from genetically rewired yeasts. Collectively, these results demonstrate that cell-based engineering strategies can activate and enhance the metabolic potential of in vitro systems to produce useful small molecules.

## Results

**Cell extracts from *S. cerevisiae* produce native and heterologous metabolites.** In this study, we aimed to advance and interweave rapid, cell-free technologies and cell-based systems that have been genetically rewired for high flux toward a particular branch of metabolism. The goal was to increase in vitro metabolic potential using value-added chemical products as a model. Before testing the impact of genetic rewiring in cells on cell-free metabolism, we first sought to activate metabolic pathways in yeast cell extracts to convert glucose to both ethanol and 2,3-butanediol (BDO) via native and heterologous enzymes, respectively. Commercial BDO is normally generated from petrochemical processes for applications in flavoring, synthetic rubber, and fuels[49,50], making it an appealing target for biosynthesis both in vivo[51,52] and in vitro[35,36,53]. Despite decades of research using yeast cell extracts to elucidate biological principles[20,22], few studies have assessed small molecule biosynthesis from heterologous pathways with *S. cerevisiae* extracts. As such, a yeast cell-free system for heterologous metabolite biosynthesis had not previously been rigorously tested or optimized. To establish such a platform, we began with an extract preparation protocol optimized for *S. cerevisiae* cell-free gene expression (CFE)[54,55]. In brief, yeast cultures were grown in 1 L flasks, washed and resuspended in buffer, lysed via high-pressure homogenization, and clarified by centrifugation (Fig. 1a). It was necessary to characterize the metabolic activity of *S. cerevisiae* cell extract produced through this method, since it was optimized for transcription and translation with catabolism only assessed in the context of ATP regeneration[56].

Both a BY4741 wildtype yeast strain expected to convert glucose to ethanol[22] and a BY4741 strain producing BDO[52] were prepared into an extract to assess metabolic activity in vitro. The BDO strain expressed *AlsD* and *AlsS* from *Bacillus subtilis* to convert pyruvate to acetoin and *NoxE* from *Lactococcus lactis* for NAD recycling[52] (Fig. 1b). The resulting cell extracts were combined with 120 mM glucose, 1 mM cofactors NAD, ATP, and CoA, as well as salts and buffer and incubated for 20 h at 30 °C. HPLC analysis confirmed that the yeast extracts retained metabolic activity for both native and heterologous pathways (Fig. 1c) with ethanol being produced from both extracts and the desired BDO product being synthesized at lower levels than ethanol, as would be expected from simple heterologous pathway complementation[52]. Nevertheless, these experiments validated that *S. cerevisiae* extracts and the conditions utilized here were suitable to produce both endogenous and heterologous metabolites.

**Cell extracts from metabolically rewired strains have increased flux toward BDO.** The optimization of CFE systems has incorporated several engineered strains for increased protein synthesis yields[27,57–59] and for specialized applications, such as producing

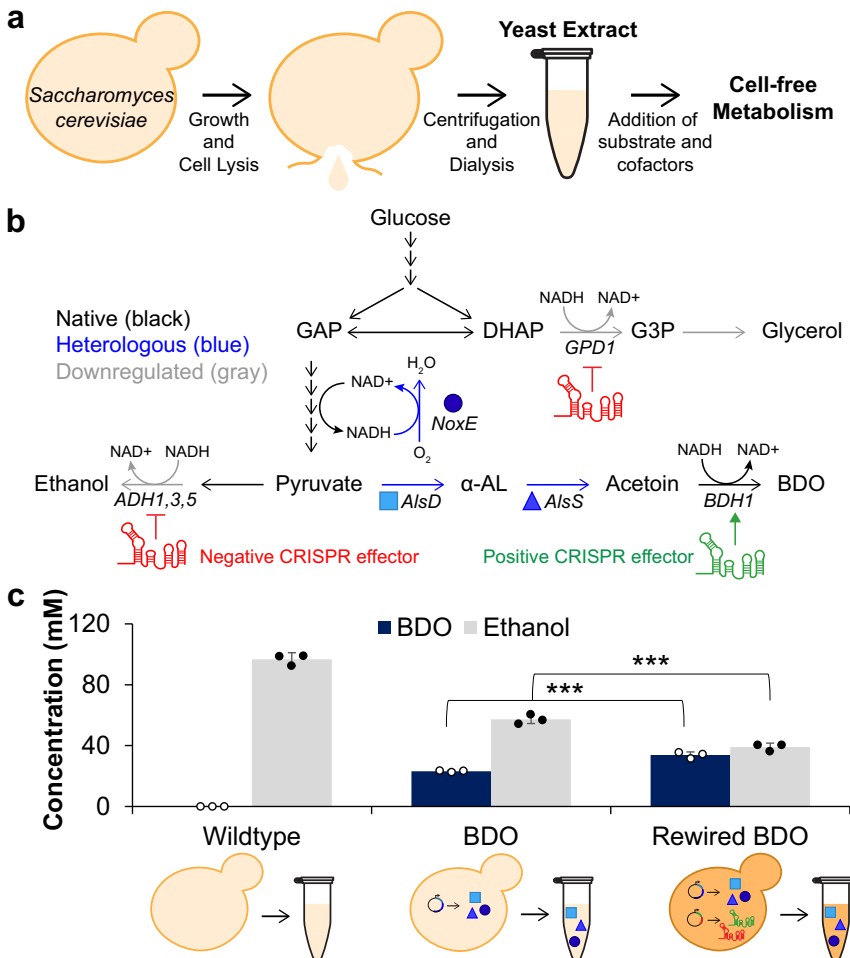

**Fig. 1 Metabolic rewiring in vivo is reflected in cell extracts. a** Schematic overview of yeast cell extract preparation for cell-free reactions. **b** Metabolic map showing wild-type yeast metabolism in black, heterologous enzymes for BDO strains in blue, and CRISPR effectors expressed by rewired BDO strain. **c** Initial cell-free comparison of wildtype, BDO, and rewired BDO cell extracts converting glucose to native and heterologous metabolites. Extract from the metabolically rewired BDO strain produces more BDO and less ethanol than extract from the unmodified BDO strain (***$p = 0.001$ as determined by a two-tailed Student's $t$-test). Data represent mean ± standard deviation of $n = 3$ technical replicates. Source data are provided as a Source Data file.

proteins with disulfide bonds[60] or noncanonical amino acids[61,62]. In contrast, nearly all cell-free, crude extract-based metabolite biosynthesis to date has relied on strains with wildtype metabolism expressing pathway enzymes[36,38,47] with few examples of strain or extract modifications to increase in vitro product titers[63,64]. Therefore, we sought to test the central hypothesis of this work with the system described above in place. Specifically, we utilized a modified *S. cerevisiae* strain that was metabolically rewired for increased BDO production using multiplexed CRISPR-dCas9 modulation[65] to simultaneously downregulate *ADH1,3,5* and *GPD1* to reduce byproduct formation while upregulating endogenous *BDH1* to increase flux to BDO (Fig. 1b)[52]. Cell-free reactions containing extract from the rewired BDO strain indeed retained altered flux to produce on average 46% more BDO and 32% less ethanol than extracts produced from the unmodified BDO strain (Fig. 1c). Moreover, although the BDO pathway itself reduces BY4741 growth rates, the additional rewiring did not further impede cell growth (Supp. Figure 1a) and thus did not extend the extract preparation protocol. Downregulation of transcription for most byproduct genes was confirmed by qPCR, and *BDH1* was slightly upregulated (Supp. Figure 1b).

While successful, we wondered whether the cell growth phase at harvest impacted overall cell-free performance. Surprisingly,

extracts obtained from cells grown to variable cell densities ($OD_{600}$ 2–8) resulted in comparable concentrations of BDO and byproducts in vitro (Supp. Figure 2b). Based on this robustness (which is atypical for cell-free protein synthesis yields[55,66]), we selected a harvest $OD_{600}$ of approximately 8 for subsequent experiments to maximize the volume of recovered extract. Despite the confirmed decrease in *GPD1* transcription after CRISPR-mediated rewiring (Supp. Figure 1b), reactions with rewired BDO extracts produced comparable glycerol titers to reactions with unmodified BDO extracts (Supp. Figure 2b). We targeted *GPD1* as the dominant, cytosolic isozyme of glycerol-3-phosphate dehydrogenase, but the mitochondrial *GPD2* isozyme could also lead to glycerol production[67]. However, simultaneously decreasing abundance of ADH and GPD enzymes leads to osmotic stress and NADH recycling deficiencies that might not be sufficiently balanced by heterologous *NoxE* expression and could, therefore, decrease flux toward BDO[68,69]. Overflow metabolism could also be difficult to overcome through transcriptional rewiring with the large bolus of glucose added to cell-free reactions. These considerations led us to use ethanol as the primary readout for byproduct formation during BDO biosynthesis. Having demonstrated the premise that in vivo metabolic engineering techniques can increase in vitro BDO biosynthesis, we sought to further improve product titers by two approaches as

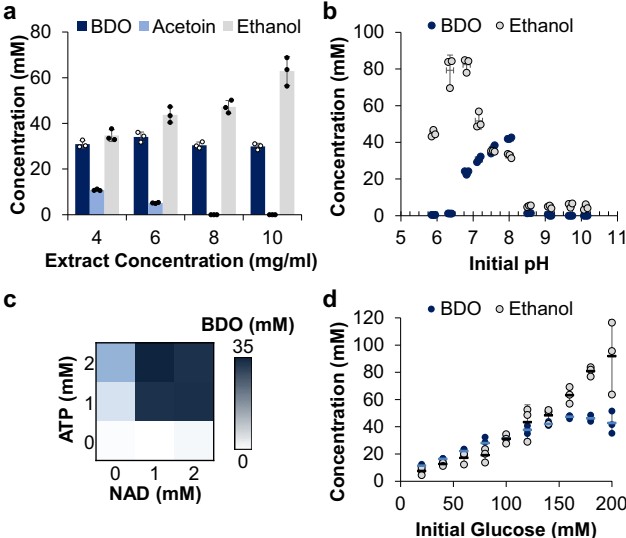

**Fig. 2 Optimizing the cell-free environment increases BDO titers. a** The concentration of rewired BDO cell extract was adjusted to maximize conversion of acetoin to BDO while minimizing ethanol production. **b** pH was varied in physiologically compatible ranges that would not be possible in vivo to highlight the disparate pH optima for BDO and ethanol biosynthesis in vitro. **c** NAD and ATP concentrations were covaried to optimize the cofactor pool. **d** Glucose concentration was varied from 20–200 mM to compare the ratio of BDO to ethanol and the final BDO titer. Data represent mean ± standard deviation of $n = 3$ technical replicates. Source data are provided as a Source Data file.

described in the following two sections: reaction optimization and strain background selection.

**Cell-free metabolic engineering (CFME) increases BDO titer.** As the first layer of optimization, we applied CFME techniques to increase BDO titers[27–29]. To do so, extract concentration, reaction pH, cofactor concentrations, and glucose concentration were sequentially optimized. Low extract concentrations resulted in the incomplete conversion of acetoin to BDO, while high extract concentrations resulted in greater ethanol production, with 6 mg yeast extract protein/mL exhibiting optimal performance (Fig. 2a). As a unique advantage of in vitro systems over in vivo hosts, it was possible to easily alter the reaction pH by directly adding acid or base, thereby revealing different pH optima for ethanol and BDO biosynthesis (Fig. 2b). From this analysis, we found that reactions started at pH 8 were more productive for BDO than those conducted at the initial condition of pH ~7.5. Similar investigations of pH impact on native and heterologous pathways in cell-free systems support this finding[39,42,46]. Next, we were able to alter the concentrations of key cofactors including NAD, ATP, and CoA. These experiments revealed that the most productive reactions contained 1 mM NAD and 2 mM ATP (Fig. 2c), whereas additional CoA did not significantly impact BDO production (as expected for a pathway without CoA-bound intermediates) (Supp. Figure 3).

Finally, we evaluated the impact of substrate concentrations on cell-free metabolism and uncovered a balance between heterologous small molecule production and overflow metabolism to native byproducts. Specifically, reactions with lower glucose levels (i.e., < 100 mM) favored BDO production and restricted ethanol whereas reactions with more elevated glucose levels saw a substantial overflow of carbon flux toward ethanol and led to decreases in BDO production (Fig. 2d). To maximize the total BDO titer, we subsequently chose 160 mM glucose as it sat at the

inflection point between BDO-favoring and ethanol-overflow metabolism regimes. Through this CFME approach, it was possible to increase BDO titer by ~40% with only small differences between initial reaction composition and final parameter selection, thus demonstrating the tunability of CFME systems (compare production in Fig. 1c relative to Fig. 2d).

**Strain background selection influences metabolic activity and rewiring.** After optimizing the physiochemical environment of cell-free reactions, we sought to evaluate whether the chassis strain selection was critical for in vitro metabolism. To do so, we utilized three strain backgrounds (BY4741, CEN.PK, and Sigma) each rewired using a multiplexed CRISPR-dCas9 approach (Supp. Table 1) and applied the optimized reaction conditions described above (6 mg/mL cell extract, pH 8, 1 mM NAD, 2 mM ATP, and 160 mM glucose). The corresponding extracts for these strains exhibited different behaviors. Cell-free reactions using the BY4741 and Sigma yeast extracts displayed similar biochemical profiles with metabolic rewiring increasing BDO production and decreasing ethanol production, whereas the CEN.PK extracts produced comparable BDO titers with and without rewiring (Fig. 3a). Notably, extract with the highest BDO titers in vitro was derived from the strain that produced the greatest BDO titer in vivo[52] (Supp. Figure 4). Specifically, reactions driven by an extract from the rewired BDO Sigma strain produced 96% more BDO and 63% less ethanol than reactions with an extract from the unmodified BDO Sigma background strain. In contrast, glycerol byproduct concentrations remained comparable across all 9 yeast extracts (Supp. Fig. 5). This effort highlights the potential to

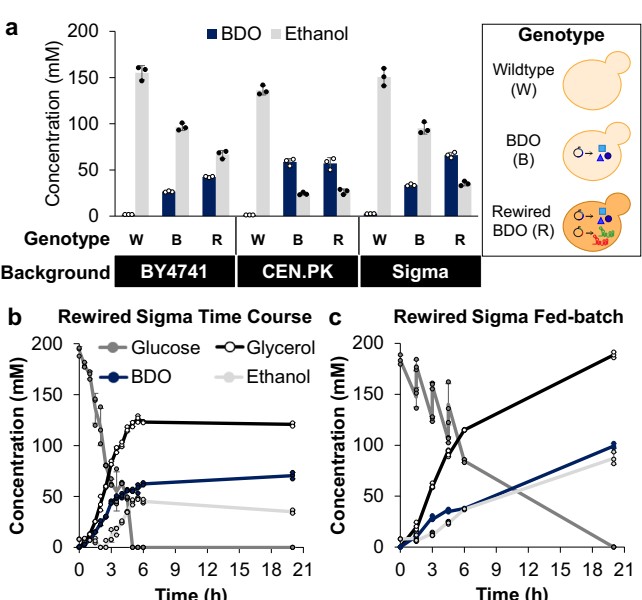

**Fig. 3 Yeast extracts from different strain backgrounds exhibit similar metabolism and rapidly consume glucose. a** Panel of cell-free reactions containing yeast extracts from 3 strain backgrounds with the optimized conditions from Fig. 2. All extracts retained metabolic activity, consuming glucose and producing ethanol and BDO. Extract from the rewired BDO Sigma strain produced the highest BDO titer and was characterized further. **b** Time course of cell-free reactions containing extract from the rewired BDO Sigma strain showing rapid glucose consumption. **c** Fed-batch reaction with additions of 45 mM glucose, 1 mM NAD, and 2 mM ATP at 1.5 h intervals. Data represent mean ± standard deviation of $n = 3$ technical replicates, and time courses represent separate samples quenched at each time point. Source data are provided as a Source Data file.

compare relative pathway performance between cell and cell-free platforms, as we have observed before[33].

Finally, to better understand the dynamics of yeast cell-free biosynthesis reactions, we conducted a time course using extract derived from the rewired Sigma strain. These reactions exhibited a substantial increase in volumetric productivity compared to the corresponding strain grown in vivo[52] with all the glucose consumed and 80% of the BDO produced within 5 h of incubation (Fig. 3b). Differences in these profiles to the in vivo instantaneous rates of BDO synthesis in the late exponential to early stationary phase (when cells were harvested for extract preparation) suggest further differences between the in vivo and in vitro kinetics beyond biomass production (Supp. Figure 4). Cell-free reactions produced BDO at $10.4 \pm 0.2$ mM/h ($0.93 \pm 0.02$ g/L-h) to reach $62.4 \pm 1.2$ mM BDO in 6 h. We next compared this rate to cellular production using empirical values to convert extract protein concentration into relative biomass from the harvested cells. One gram of cells produces ~1 mL of extract, the rewired BDO Sigma extract contained $27 \pm 3$ mg protein per mL (Supp. Table 2), and optimized reactions contained 6 mg protein per mL. Combining these values results in $216 \pm 28$ mg cell biomass/mL cell-free reaction, which includes both unlysed cells and insoluble components lost during centrifugation. Comparing BDO synthesis in vivo and in vitro over time (Supp. Table 3) indicates that the cell-free system produces BDO more rapidly than cellular fermentation (Supp. Table 4). However, at the point where product formation stops or significantly slows, (~6 h in vitro and 48 h in vivo), the mass normalized productivity is comparable between the two systems ($0.055 \pm 0.0038$ mmol BDO/h/g cells in vivo and $0.048 \pm 0.0062$ mmol BDO/h/g cells in vitro) (Supp. Table 4). While similar, direct comparisons are difficult due to the number of assumptions involved. For example, our analysis here only accounts for the time between glucose addition and product analysis (i.e., the biochemical conversion in vivo or in vitro). Extract preparation adds additional time to the overall cell-free process, but product formation still occurs during cell growth prior to lysis. Nevertheless, genetic rewiring that serves to increase desired metabolite production in vivo likewise improves production in vitro.

Based on these rapid cell-free reaction rates, we hypothesized that a fed-batch approach could further maximize BDO titer while maintaining relatively short reaction times[36]. To accomplish this, it was necessary to increase NAD and ATP supply to enable full glucose utilization with fed-batch spiking (Supp. Fig. 6a, b). Ultimately, fed-batch reactions receiving additional glucose, NAD, and ATP (45, 1, and 2 mM, respectively) at 1.5 h intervals displayed a reduced rate of BDO synthesis, but a higher overall titer of $99.3 \pm 2.3$ mM (Fig. 3c). The decreased yield observed in this fed-batch system and the requirement of cofactor spiking suggest limitations due to cofactor recycling and/or overflow metabolism, which could potentially be alleviated by increasing NoxE expression or reducing the magnitude of glucose spikes through a continuous feed[70].

**Extending the integrated cellular/cell-free metabolic engineering approach to alternative metabolites**. After demonstrating the impact of using genetically rewired strains to increase flux for cell-free BDO production, we assessed whether the integrated in vivo metabolic rewiring and in vitro biochemical transformation approach was generalizable for alternate products. Specifically, we sought to increase the biological production of itaconic acid, a versatile biochemical precursor for plastics and resins[71,72], and glycerol, a ubiquitous additive in commodity chemicals, food products, and industrial sectors[73,74].

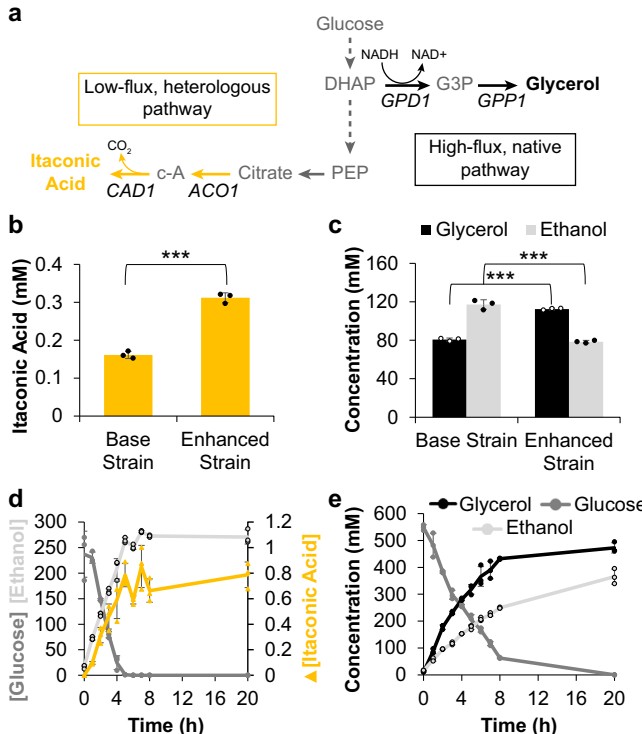

**Fig. 4 Cell-free reactions with extract from engineered yeast strains enable production of diverse metabolites. a** Metabolic map showing the conversion of glucose to glycerol via dihydroxyacetone phosphate (DHAP) and glycerol-3-phosphate (G3P) or itaconic acid via citrate and cis-aconitate (c-A). **b, c** Initial cell-free reaction conditions confirm that strains engineered for increased product titers result in cell extracts with greater flux toward products of interest (***$p \le 0.001$ compared to base strain). From a two-tailed Student's $t$-test, $p = 8.45 \times 10^{-5}$ for itaconic acid, $p = 7.16 \times 10^{-6}$ for glycerol, and $p = 2 \times 10^{-3}$ for ethanol. **d-e** Yeast cell-free reactions with optimized conditions rapidly convert glucose into products, resulting in comparable concentrations to in vivo efforts in significantly less time. Data represent mean ± standard deviation of $n = 3$ technical replicates, and time courses represent separate samples quenched at each time point. Source data are provided as a Source Data file.

The selection of itaconic acid and glycerol can demonstrate yeast CFME for a low-flux, heterologous pathway, and a high-flux, native pathway, respectively (Fig. 4a). The base strain for itaconic acid production expressed the cis-aconitate decarboxylase gene, CAD1, from Aspergillus terreus, and the enhanced strain expressed this gene in the background with genomic knockouts of ade3 and bna2[72]. The base strain for glycerol production contained an empty plasmid backbone, while the enhanced strain contained a plasmid to overexpress GPD1 and GPP1[74]. Extracts from base strains and metabolically rewired strains were established for both products using methods developed above (Supp. Tables 1–2). An initial pair of glycerol strains was generated through CRISPR-mediated metabolic rewiring with guide RNAs targeting positive effectors to upregulate GPD1 and GPP1[74], and the resulting extract from the rewired strain produced 18.7% more glycerol than extract from the control strain (Supp. Fig. 7a). This demonstrates the generalizability of in vivo CRISPR-dCas9 rewiring to increase in vitro biosynthesis, but these extracts were unable to consume high concentrations of glucose. For this reason, we carried out further glycerol biosynthesis reactions with extracts from the plasmid-rewired strains to maximize cell-free glycerol titers.

Initial reaction conditions reinforce the observations from the BDO extract optimizations that metabolic rewiring in vivo transfers to cell extracts with similarly altered flux. Cell-free reactions containing extract from the enhanced strains produced 94% more itaconic acid and 39% more glycerol than the corresponding reactions with an extract from base strains (Fig. 4b, c). Optimization of reaction conditions further boosted these titers by more than 3-fold (Supp. Figs. 7b, 8b) with tuned reaction conditions producing up to $0.85 \pm 0.17$ mM itaconic acid in 7 h from 240 mM glucose (Fig. 4d) and up to $472.4 \pm 19.3$ mM glycerol in 20 h from 500 mM glucose (Fig. 4e). Similar to BDO biosynthesis, the in vitro platform enables relatively high volumetric productivities of $0.12 \pm 0.02$ mM/h ($15.9 \pm 3.0$ mg/L-h) itaconic acid and $54 \pm 0.28$ mM/h ($4.98 \pm 0.03$ g/L-h) glycerol in the first 8 h when considering only the time of in vitro bioconversion, since glucose is also converted to the product during cell growth. As a means of studying metabolic limitations, we observed that reactions with an extract from the enhanced strain also produced more itaconic acid than extract from the base strain when provided with citrate instead of glucose as the substrate, suggesting that flux limitations likely stem from the tricarboxylic acid cycle (Supp. Fig. 8a).

## Discussion

In this study, we developed an integrated in vivo/in vitro metabolic engineering approach to activate cell-free metabolism by genetically modulating metabolic pathways in yeast source strains used for extract generation. Notably, we found that metabolic rewiring in these strains improves desired productivities in vitro. This reflects previous efforts to engineer strains for CFE to increase protein yields[27,57,58] and enable specialized applications[60–62] through gene knockouts and/or complementation, but the focus here was on reshaping metabolic flux for small molecule synthesis rather than translation. To our knowledge, this is the first report using multiplexed dCas9 modulation to enhance cell-free biosynthesis, using multiple gRNAs that can be implemented and adjusted more rapidly than knocking out enzymes in vivo[63] or selectively removing them in vitro[64]. Our approach has several key features.

First, we found that rewiring source strains used in extract generation coupled to physiochemical optimization lead to significant increases in volumetric productivities. Specifically, we achieved productivities of $10.4 \pm 0.2$ mM/h for BDO, $0.12 \pm 0.02$ mM/h for itaconic acid, and $54 \pm 0.28$ mM/h for glycerol in cell-free biosynthesis reactions, representing increases up to 3-fold above unmodified extracts. Further process optimization and analysis will be required to precisely determine and improve normalized productivities based on the cellular biomass used to generate extracts, but it is worth noting that the product formation reported is product synthesized beyond the biosynthesis that occurs in vivo prior to extracting preparation. This means that cell-free reactions can potentially be used to extend the useful lifespan of cell cultures for metabolite production without the contamination risk or selective pressures associated with continuous culture systems. Fine-tuning the metabolic landscape both before and after cell lysis improves control over biosynthesis by increasing the number of levers available for engineering.

Second, the integrated in vivo/in vitro framework demonstrates the potential for *S. cerevisiae* extracts in metabolite synthesis. Since Buchner's original work in yeast extracts led to a Nobel Prize[22,23], cell-free biosynthesis efforts in crude extracts have primarily focused on *E. coli*-based systems[27,33,34,36,38] due to the extensive history of optimization for CFE with this bacterium[24]. Although yeast has been critical to the development of industrial biotechnology, relatively low batch CFE yields[56,75] have limited

applications of yeast extracts primarily to endogenous metabolic pathways[44–46]. The methodology developed here results in highly productive cell-free reactions, and extending this approach to the biosynthesis of other products could enable new opportunities for studying fundamental metabolic bottlenecks and assessing auxiliary enzymes or cofactors for pathway optimization. With these biochemical analyses in mind, coupling the in vivo/in vitro approach with the immense library of engineered yeast strains from academic and industrial labs could catalyze breakthroughs in the biological production of diverse commodity chemicals and medically relevant compounds[3,12–14].

Third, the rewiring and lysis approach is generalizable. We successfully applied our framework to multiple base strains of *S. cerevisiae* (i.e., BY4741, CEN.PK, Sigma) for rapid biosynthesis of native and heterologous chemical products from 3 distinct branches of metabolism as a proof of concept. The broad applicability of CRISPR/dCas9 modulation across species[76,77] and the growing repertoire of protocols for cell extract preparation from diverse organisms[59,75,78] present an immense application space of rewired metabolism in cell-free contexts. This could include increasing the productivity of established *E. coli* CFME platforms for high-value molecules[30,79,80] and materials[38,81] with the potential for scaling up cell-free reactions[31,82]. Furthermore, this workflow could lead to robust cell-free systems comprising metabolically rewired extracts from species with appealing metabolic capabilities but low-yielding or undeveloped CFE platforms, such as *Yarrowia lipolytica*[83], *Pichia pastoris*[84,85], and *Streptomyces spp*[86–90], increasing opportunities for cell-free prototyping and biomanufacturing applications.

Looking forward, we anticipate that our in vivo/in vitro approach to combine cellular metabolic engineering strategies for flux rewiring with cell-free reaction optimization will enable new directions in cell-free synthetic biology.

## Methods

**Strains and media**. Strains were grown in yeast synthetic complete (YSC) media containing 1.71 g/L yeast nitrogen base (Sunrise Science Products), 5 g/L ammonium sulfate, 20 g/L glucose, and a complete supplement mixture (CSM) with the appropriate dropout(s) to enforce plasmid retention (Sunrise Science Products) according to product specifications. All strains used in this study are described in Supp. Table 1.

**Metabolic rewiring via CRISPR**. The details of CRISPR rewiring for *S. cerevisiae* BDO strains were previously described[52]. Overall, dCas9-VPR is expressed as a dual-mode regulator under the control of TDH3 promoter on a p415 Mumberg plasmid. On the same plasmid, a single guide RNA (sgRNA) cassette including sgRNAs targeting *ADH1*, *ADH3*, *ADH5*, and *GPD1* at ORFs and *BDH1* at the promoter region is expressed under the control of TEF1 promoter. gRNA sequences are provided in Supp. Table 5. The design of tRNA-sgRNA bricks is applied in the sgRNA cassette (Supp. Table 6). After being expressed, tRNAs will be cut by RNase P and Z to release the processed sgRNAs. For the control strains, a p415 plasmid containing the same TDH3-dCas9-VPR but no sgRNA cassette is used.

The details of CRISPR rewiring for *S. cerevisiae* glycerol strains were previously described[74]. Overall, dCas9-VPR is expressed as a gene activator under the control of TDH3 promoter on a p415 Mumberg plasmid. On the same plasmid, two individual sgRNAs targeting *GPD1* and *GPP1* at promoter regions are expressed under the control of SNR52 promoters. gRNA sequences are provided in Supp. Table 5. For the control strains, a p415 plasmid containing the same TDH3-dCas9-VPR but no sgRNAs is used.

**Quantitative PCR**. Quantitative PCR (qPCR) was performed to measure the expression of genes that are targeted by the CRISPR/dCas9 modulation in both rewired and control BY4741-BDO strains. Yeast cultures in biological triplicates were inoculated at OD = 0.05 and grown for 19 h at 30 °C to mid-log phase (OD = 0.9–1.2). RNA was extracted using Zymolyase digestion of the yeast cell wall followed by purification using the Quick-RNA MiniPrep kit according to the manufacturer's instructions (Zymo Research Corp). cDNA was generated from the purified RNA using the Protoscript II Reverse Transcriptase (NEB) with 20 µM d (T)23VN primer. For qPCR experiments, primers for *ADH1, ADH3, ADH5, GPD1, BDH1, and ALG9* (Supp. Table 7) were designed using the PrimerBlast tool[91] and obtained from Integrated DNA Technologies. Quantitative PCR was performed on a ViiA7 qPCR system (Life Technologies) using SYBR Green Master Mix (Applied

Biosystems) with ROX as a passive reference. qPCR was performed according to the manufacturer's instructions with an annealing temperature of 60 °C and 50 ng of total cDNA product per 10 μL reaction in a 384-well optical plate. All reactions were run in technical triplicates. The data from two biological replicates of rewired strains and three biological replicates of control strains were analyzed. The *ALG9* gene was used as a housekeeping gene. The gene expression between rewired and control strains was compared by calculating the difference in the threshold cycle ($\triangle C_t$) between the individual target genes and ALG9. Error bars represent the standard deviation of $\triangle C_t$ from the biological replicates, and p-values are derived from two-tailed Student's *t*-tests.

**Buffers and chemicals**. Buffer A contained 30 mM 4-(2-hydroxyethyl)-1-piper-azineethanesulfonic acid (HEPES) adjusted to pH 7.4 with potassium hydroxide, 100 mM potassium glutamate, 2 mM magnesium glutamate, and 2 mM dithio-threitol (DTT). Mannitol Buffer A was identical to Buffer A with the addition of 8.5% (w/v) D-mannitol. Concentrated DTT was added immediately prior to the use of each buffer. All chemicals were purchased from Sigma-Aldrich unless otherwise noted.

**Cell extract preparation**. Yeast extracts were prepared based on a method modified from previous studies in cell-free protein synthesis[54,55]. Starter cultures were grown from glycerol stocks to saturation at 30 °C and 250 rpm. These starter cultures were used to inoculate 1 L of appropriate CSM dropout media at $OD_{600}$ 0.1–0.2 in 2.5 L Tunair™ shake flasks (Millipore Sigma) as determined by mea-surement on a NanoDrop 2000c (ThermoFisher Scientific). Cells were grown to exponential phase ($OD_{600}$ considerations discussed above), and then cultures were transferred to 1 L centrifuge bottles and pelleted with a 10-min spin at $4000 \times g$ in a rotor cooled to 4 °C. The supernatant was discarded, and each cell pellet was divided into two 50 mL conical tubes. Pellets were resuspended with 10 mL ice-cold Mannitol Buffer A per conical tube and pelleted with a 5 min spin at $4000 \times g$ in a rotor cooled to 4 °C to wash the cells. The supernatant was discarded, and this wash step was repeated 2 more times for a total of 3 washes. Cell suspensions were combined for the final spin to generate a single cell pellet per strain harvested, and cell pellets were weighed prior to flash-freezing in liquid nitrogen.

Frozen cell pellets were thawed on ice for 1 h and resuspended with 1 mL Mannitol Buffer A per gram cell mass. These suspensions were lysed with one pass through an EmulsiFlex-B15 homogenizer (Avestin) at ~90 psi air pressure (26,000–28,000 psi homogenization pressure). Cellular debris was removed with a 5 min spin at $20,000 \times g$, and the supernatant was collected for a second spin to remove residual debris. The supernatant from the second spin was transferred to a 3.5 kDa Slide-a-Lyzer™ dialysis cassette (ThermoFisher) and dialyzed for 2 h at 4 °C in 1 L Buffer A with 0.5 mM phenylmethylsulfonyl fluoride (PMSF) and a single buffer exchange after the first hour. Dialyzed extracts were centrifuged again for 5 min at $20,000 \times g$ to remove precipitated proteins. The supernatant was aliquoted and flash-frozen in liquid nitrogen, and extracts were stored at −80 °C until use. The protein content of extracts was determined using a Bradford assay for technical triplicates of 4000-fold, 7000-fold, and 10,000-fold dilutions with bovine serum albumin standards using a BioTek Gen5 microplate reader. All extracts used in this study are described in Supp. Table 2.

**Cell-free biosynthesis reactions**. 20 μL reactions were set up in 1.5 mL tubes and initially run for 20 h in triplicates. Initial reactions for BDO production contained 8 mM magnesium glutamate, 10 mM ammonium glutamate, 134 mM potassium glutamate, 120 mM glucose, 100 mM BisTris buffer, 1 mM adenosine triphosphate (ATP), 1 mM nicotine adenine dinucleotide (NAD), 1 mM coenzyme A (CoA), and 8 mg/mL yeast extract. These parameters were optimized for BDO production in Fig. 3. pH of these reactions was adjusted using glacial acetic acid or potassium hydroxide prior to the addition of yeast extract and cofactors, and initial pH was measured after the addition of all reaction components using an Orion™ ROSS Ultra™ Refillable pH/ATC Triode™ (Thermo Scientific). Initial reactions for itaconic acid and glycerol production contained the same concentrations of salts, glucose, buffer, and CoA as the initial BDO reactions. However, NAD and ATP were increased to 3 mM to facilitate glucose consumption prior to cofactor optimization, and extract concentration was set at 6 mg/ml.

**In vivo measurements**. Cultures of BDO-producing *S. cerevisiae* strains were inoculated into 25 mL of YSC media within 250 mL shake flasks (PYREX) at $OD_{600}$ 0.01 in biological triplicates and cultivated at 30 °C. Corresponding cultures were started at three different time points (0, 8, and 14 h) to accomplish 24h-sampling. 1 mL samples were taken intermittently and centrifuged at 16,000 g. The super-natant was filtered with a 0.2 μm sterile syringe filter (VWR International) prior to high-performance liquid chromatography (HPLC) analysis for the production of BDO and the consumption of glucose. Samples were separated using an HPLC Ultimate 3000 (Dionex) equipped with an Aminex HPX-87H ion exclusion column (BioRad) and analyzed using the Chromeleon 7.2 Chromatography Data System (Dionex). A 10 μL injection volume was used in an isocratic mobile phase of 5 mM $H_2SO_4$ (pH = 2) at a flow rate of 0.6 mL/min. The column temperature was 60 °C and a refractive index detector (RID) was used at a temperature of 25 °C. A 50 g/L standard stock was made from 98% 2,3-butanediol (Sigma) and serially diluted

from 20 g/L to 0.625 g/L to make BDO standards. A 40 g/L standard stock was made from D-(+)-Glucose Anhydrous (MP Biomedicals) and serially diluted from 20 g/L to 0.625 g/L to make glucose standards.

**In vitro Metabolite analysis**. Reactions were quenched with 20 μL of 10% (w/v) trichloroacetic acid, and precipitated proteins were removed by centrifugation for 10 min at $20,000 \times g$. The supernatant was transferred to vials and 5 μL were injected on an Agilent 1260 HPLC system. Metabolites were separated with 5 mM sulfuric acid flowing at 0.6 mL/min on a Rezex™ ROA-Organic Acid H+ (8%) LC Column (Aminex) at 20 °C. Metabolite concentrations were determined using a refractive index detector (RID) or a diode array detector (DAD) at 210 nm based on the retention time of standard solutions for each compound. Cell-free reactions were performed in triplicates, and error bars on all figures represent the standard deviation of 3 technical replicates. Significant differences between conditions were determined with a two-tailed Student's *t*-test, and time courses were completed with separate samples for each time point prepared from a single master mix. Data were analyzed and visualized using Agilent ChemStation and Microsoft Excel prior to assembly with graphics in Adobe Illustrator.

**Reporting summary**. Further information on research design is available in the Nature Research Reporting Summary linked to this article.

## Data availability
All data generated and analyzed for this study are presented in the manuscript or supplementary information or are available upon request from the corresponding author. There are no restrictions on data availability, and the source data are provided with this publication. Source data are provided with this paper.

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

## Acknowledgements

We thank Ashty Karim and Bastian Vögeli for helpful discussions regarding experiments and figures for this work. In addition, we thank Nigel Mouncey, Yasuo Yoshikuni, Susannah Tringe, and Rex Malmstrom from the U.S. Department of Energy Joint Genome Institute for conversations throughout the development of the project. This work was supported by an Emerging Technologies Opportunity Program (ETOP) award under Subcontract No. 7399340 from the U.S. Department of Energy Joint Genome Institute, a DOE Office of Science User Facility, which is supported by the Office of Science of the U.S. Department of Energy under Contract No. DE-AC02-05CH11231. M. C.J. gratefully acknowledges the David and Lucile Packard Foundation and the Camille Dreyfus Teacher–Scholar Program. B.J.R. is supported by a National Defense Science and Engineering Graduate Fellowship (Award ND-CEN-017-095).

## Author contributions

B.J.R. and H.B. performed in vitro experiments and analyzed data. X.Y. collected and analyzed in vivo data. H.S.A. and M.C.J. performed supervisory roles. All authors conceived of experiments and contributed to writing and editing the paper.

## Competing interests

The authors declare no competing interests.
