## [Peer Review File · Nature Communications]

Reviewers' Comments:

Reviewer #1:

Remarks to the Author:

This work by Rasor et al. is a very well-written study and useful addition to the cell free literature, especially the growing body of work on cell free metabolic engineering. It describes extract preparation for Cas9 modified yeast extracts and shows the utility for screening enzymes and cell type on yield of small molecules (butanediol, itaconic acid, and glycerol). This publication is suitable for publication after a few minor edits:

1. The statement on Pg. 10, lines 247 to 250 reads as follows:

"While the continuously changing biomass concentration of cell cultures precludes a normalized comparison to cell-free reactions with static protein content, a holistic comparison of the two systems reveals a noteworthy difference in overall volumetric productivity."

While the cells do change in concentration during in vivo expression, there is an end biomass to which you can and should normalize. By only providing volumetric comparison it appears that the cell free production is more efficient, which is likely not the case on a per mass (or per cell unit) basis. Mass normalization should be included alongside the volumetric.

2. In this same section, time of production is also compared. They authors must be careful on this point, as the cell free reaction did only take 6 hours, but how long did it take to grow and process the cell extract? It is not a direct comparison. Either a clarifying statement on time for extract prep must be added in this section or this direct time comparison should not be made.

3. Regarding methods, details on CRISPR technique would be helpful. The technique is mentioned and the reference is cited but there is no CRISPR subsection in the Methods section of the manuscript. Is the cited literature sufficient for learning/repeating this technique? It helps to provide some details in this manuscript as well, such as did you express the CRISPR proteins and guide RNAs in-house or did you need to purchase them? What vendor?

Reviewer #2:

Remarks to the Author:

In the manuscript titled "An integrated in vivo/in vitro framework to enhance cell-free biosynthesis with metabolically rewired yeast extracts" Rasor et al. describe cell free systems of yeast strains useful for production of important metabolites. The authors demonstrate yeast strains engineered for 2,3-butanediol (BDO) production outperform WT yeast strains. The authors show, that strains rewired via CRISPR interference/activation are capable of reaching high volumetric productivities. The authors systematically analyze the effect of cofactors and reaction pH to optimize BDO production. Importantly, at times, cell free systems outperform cell based systems. Subsequently the authors also demonstrate cell free systems of yeast strains engineered for production of itaconic acid and glycerol have high productivities compared to wild type extracts.

Overall the work is technically sound and well conducted. The most impactful feature of the manuscript is the demonstration that CRISPRi/a rewiring of metabolism coupled with cell free systems. This is important as it can potentially expedite the cycle of design-build-test for cell free systems, however this should be demonstrated for itaconic acid and/or glycerol production also. (See comment below)

Major:

The authors describe the CRISPRi/a scheme in figure 1b for the BDO rewired strain and show inhibition of GPD1. However, supplemental figure 1c shows glycerol levels are unaffected between WT, BDO and BDO rewired. This is counter intuitive; can the authors explain this? To this end, the authors should validate that each guide is inhibiting or activating transcription at respective target loci (such as via quantitative PCR). It is possible that some guides contribute more to BDO rewired productivity than others, however this should be quantified to understand the 'rewiring' that has

occurred.

The authors present a framework for CRISPRi/a rewiring coupled with cell free systems however they only provide the one example of BDO production. It is possible that BDO is unique and CRISPRi/a optimization is not as effective for other pathways in cell free systems. For instance, the authors should apply a similar CRISPRi/a rewiring methodology to itaconic acid and/or glycerol production.

Minor:

The authors should provide a supplemental table with guide RNA sequences. This is important for thorough documentation.

Response to reviewers (responses highlighted in blue):

Reviewer #1:

This work by Rasor et al. is a very well-written study and useful addition to the cell free literature, especially the growing body of work on cell free metabolic engineering. It describes extract preparation for Cas9 modified yeast extracts and shows the utility for screening enzymes and cell type on yield of small molecules (butanediol, itaconic acid, and glycerol). This publication is suitable for publication after a few minor edits:

We appreciate the positive feedback from the reviewer regarding the quality of the manuscript and results.

1. The statement on Pg. 10, lines 247 to 250 reads as follows: “While the continuously changing biomass concentration of cell cultures precludes a normalized comparison to cell-free reactions with static protein content, a holistic comparison of the two systems reveals a noteworthy difference in overall volumetric productivity.”

While the cells do change in concentration during *in vivo* expression, there is an end biomass to which you can and should normalize. By only providing volumetric comparison it appears that the cell free production is more efficient, which is likely not the case on a per mass (or per cell unit) basis. Mass normalization should be included alongside the volumetric.

We thank the reviewer for raising this point of clarification.

In the original manuscript, we tried to avoid strong comparisons between *in vitro* and *in vivo* systems because we feel that there is no easy way to compare them in an apples-to-apples fashion. For example, from a process perspective, one would need to consider reactor space, time for cell growth, media, harvest OD, and extract production in addition to the *in vitro* reaction's yields and titers on glucose. Many of these variables were not optimized and there are likely many trade-offs. Another wrinkle for making direct comparisons was that we previously took the best reported titers from the literature using the same strains, which did not provide sufficient information for mass normalization.

That said, to address the reviewer's question and avoid overstatements, we carried out additional experiments so that we could include mass normalization alongside the volumetric rates. We focused on BDO production because it was our primary case study and applied an empirical approach to limit the necessary assumptions. For the *in vivo* system, we measured biomass and metabolites for cell cultures in triplicate over 72 hours, noting that product formation in cells ceased at 48 hours. For the cell-free system, we used a basis of 1 gram of harvested cell biomass per 1 mL of extract (i.e., 1000 mg biomass / ml extract) and divided this value by the protein content of the best performing extract, rewired BDO Sigma (27.74 ± 3.48 mg protein / mL as reported in Supp. Table 2). This provides a conversion factor of ~36 mg cell biomass per mg extract protein, which

can be multiplied by 6 mg extract per mL cell-free reaction to reach ~216 mg cell biomass per mL cell-free reaction. Although this does not account for incomplete cell lysis or other variable factors, we believe it enables a suitable and conservative comparison of normalized productivities between these two systems. Metabolite and biomass data are reported in the new Supplementary Table 3, and we used these data to normalize volumetric productivities by cellular biomass in Supplementary Table 4.

Supplementary Table 3. Biomass and metabolite concentrations in cultures or cell-free reactions over time. Values show mean \pm standard deviations of three ($n = 3$) independent experiments (biological replicates *in vivo* and technical replicates *in vitro*). Cell-free reactions contained 6 mg extract protein per mL, which corresponds to ~216 mg cell biomass per ml for this extract (1 g biomass yields ~1 mL of cell extract, and the rewired BDO Sigma extract contained 27.74 ± 3.48 mg protein / ml). Bolded rows represent the time at which significant product formation ceases in each system.

System	Time point (h)	Cell biomass (mg/ml)	BDO (mM)
In vivo	0	0.0407 ± 0	0 ± 0
In vivo	24	5.45 ± 0.22	10.37 ± 0.71
In vivo	48	9.42 ± 0.44	24.79 ± 1.28
In vivo	72	9.82 ± 0.44	24.95 ± 0.45
Cell-free	0	216 ± 27.57	0.57 ± 0.07
Cell-free	6	216 ± 27.57	62.43 ± 1.24
Cell-free	20	216 ± 27.57	70.46 ± 3.12

Supplementary Table 4. Volumetric productivities normalized to cellular biomass. Values show mean \pm propagated error from three independent experiments. Bolded rows represent the time at which significant product formation ceases in each system.

System	Time point (h)	mM BDO / h	mmol BDO / h / g cell biomass
In vivo	24	0.432 ± 0	0.079 ± 0.0063
In vivo	48	0.516 ± 0.015	0.055 ± 0.0038
In vivo	72	0.346 ± 0.018	0.035 ± 0.0017
Cell-free	6	10.41 ± 0.011	0.048 ± 0.0062
Cell-free	20	3.523 ± 0.16	0.017 ± 0.0022

Our results indicate that the normalized yield when product formation stopped or significantly slowed (6 hours *in vitro* and 48 hours *in vivo*) is 0.048 ± 0.0062 mmol BDO / h / g cells in the cell-free system and 0.055 ± 0.0038 mmol BDO / h / g cells *in vivo*. These data, which are included on page 11 of the revised manuscript, show that the normalized yield is comparable for the cell-free system and cells in this case and context. Of note, given that we now provide our own *in vivo* data for BDO, we removed literature comparisons to cellular glycerol and itaconic acid production as well.

In the revised manuscript, we chose to simply report these data rather than highlight them strongly due to the assumptions involved in estimating the cellular biomass represented

by a cell extract. Looking forward, we hope to carefully explore these calculations in a future manuscript with a more challenging pathway and complete process optimization. That said, we reiterate that the goal of this manuscript was to demonstrate an integrated *in vivo/in vitro* framework to enhance cell-free biosynthesis with metabolically rewired yeast extracts. This goal was achieved, which we believe will expand opportunities for synthetic biology in accelerating biosynthetic pathway prospecting and rapid prototyping of metabolism.

2. In this same section, time of production is also compared. They authors must be careful on this point, as the cell free reaction did only take 6 hours, but how long did it take to grow and process the cell extract? It is not a direct comparison. Either a clarifying statement on time for extract prep must be added in this section or this direct time comparison should not be made.

The reviewer is correct in pointing out that the overall process for cell-free biosynthesis is longer than an individual reaction. As stated above, it is difficult to directly compare the cell-free and *in vivo* system. In the revised manuscript, we have provided several clarifying statements per the reviewer's request.

Page 12 – “Note that this comparison only accounts for the time between glucose addition and product analysis (i.e., the biochemical conversion *in vivo* or *in vitro*). Extract preparation adds additional time to the overall cell-free process, but product formation still occurs during cell growth prior to lysis.”

Page 14 – “Similar to BDO biosynthesis, the *in vitro* platform enables relatively high volumetric productivities of 0.12 ± 0.02 mM/h (15.9 ± 3.0 mg/L-h) itaconic acid and 54 ± 0.28 mM/h (4.98 ± 0.03 g/L-h) glycerol in the first 8 h when considering only the time of *in vitro* bioconversion, since glucose is also converted to product during cell growth.”

Page 17 – “Further process optimization and analysis will be required to precisely determine and improve normalized productivities based on the cellular biomass used to generate extracts, but it is worth noting that the product formation reported is product synthesized beyond the biosynthesis that occurs *in vivo* prior to extract preparation. This means that cell-free reactions can potentially be used to extend the useful lifespan of cell cultures for metabolite production without the contamination risk or selective pressures associated with continuous culture systems.”

3. Regarding methods, details on CRISPR technique would be helpful. The technique is mentioned and the reference is cited but there is no CRISPR subsection in the Methods section of the manuscript. Is the cited literature sufficient for learning/repeating this technique? It helps to provide some details in this manuscript as well, such as did you express the CRISPR proteins and guide RNAs in-house or did you need to purchase them? What vendor?

We agree with the reviewer that additional details on the CRISPR technique would be helpful. As such, an additional subsection on metabolic rewiring via CRISPR was included in the methods to elaborate these steps in greater detail.

“Rewiring for BDO Strains

The details of CRISPR rewiring for *S. cerevisiae* BDO strains were previously described¹. In brief, dCas9-VPR is expressed as a dual-mode regulator under the control of TDH3 promoter on a p415 Mumberg plasmid. On the same plasmid, a single guide RNA (sgRNA) cassette including sgRNAs targeting *adh1*, *adh3*, *adh5*, and *gpd1* at ORFs and *BDH1* at the promoter region is expressed under the control of TEF1 promoter. gRNA sequences are provided in **Supp. Table 5**. The design of tRNA-sgRNA bricks is applied in the sgRNA cassette (**Supp. Table 6**). After being expressed, tRNAs will be cut by RNase P and Z to release the processed sgRNAs. For the control strains, a p415 plasmid containing the same TDH3-dCas9VPR but no sgRNA cassette is used.

Rewiring for Glycerol Strains

The details of CRISPR rewiring for *S. cerevisiae* glycerol strains were previously described². Overall, dCas9VPR is expressed as a gene activator under the control of TDH3 promoter on a p415 Mumberg plasmid. On the same plasmid, two individual sgRNAs targeting *gpd1* and *gpp1* at promoter regions are expressed under the control of SNR52 promoters. gRNA sequences are provided in **Supp. Table 5**. For the control strains, a p415 plasmid containing the same TDH3-dCas9-VPR but no sgRNAs is used.

Quantitative PCR Protocol and Analysis for Targets Validation

Quantitative PCR (qPCR) was performed to measure the expression of genes that are targeted by the CRISPR/dCas9 modulation in both rewired and control BY4741-BDO strains. Yeast cultures in biological triplicates were inoculated at OD=0.05 and grown for 19h at 30°C to mid-log phase (OD = 0.9 - 1.2). RNA was extracted using Zymolyase digestion of the yeast cell wall followed by purification using the Quick-RNA MiniPrep kit according to manufacturer's instructions (Zymo Research Corp). cDNA was generated from the purified RNA using the Protoscript II Reverse Transcriptase (NEB) with 20μM d(T)23VN primer. For qPCR experiments, primers for ADH1, ADH3, ADH5, GPD1, BDH1, and ALG9 (**Supp. Table 7**) were designed using the PrimerBlast tool and obtained from Integrated DNA Technologies. Quantitative PCR was performed on a ViiA7 qPCR system (Life Technologies) using SYBR Green Master Mix (Applied Biosystems) with ROX as a passive reference. qPCR was performed according to manufacturer's instructions with an annealing temperature of 60°C and 50 ng of total cDNA product per 10 μL reaction in a 384-well optical plate. All reactions were run in technical triplicates. The data from two biological replicates of rewired strains and three biological replicates of control strains was analyzed. The ALG9 gene was used as a housekeeping gene. The gene expression between rewired and control strains was compared by calculating the difference in the threshold cycle (ΔC_t) between individual target gene and ALG9. Error bars represent the standard deviation of ΔC_t from the biological replicates, and p-values are derived from two-tailed Student's t-Tests.”

Reviewer #2:

In the manuscript titled “An integrated in vivo/in vitro framework to enhance cell-free biosynthesis with metabolically rewired yeast extracts” Rasor et al. describe cell free systems of yeast strains useful for production of important metabolites. The authors demonstrate yeast strains engineered for 2,3-butanediol (BDO) production outperform WT yeast strains. The authors show, that strains rewired via CRISPR interference/activation are capable of reaching high volumetric productivities. The authors systematically analyze the effect of cofactors and reaction pH to optimize BDO production. Importantly, at times, cell free systems outperform cell based systems. Subsequently the authors also demonstrate cell free systems of yeast strains engineered for production of itaconic acid and glycerol have high productivities compared to wild type extracts.

Overall the work is technically sound and well conducted. The most impactful feature of the manuscript is the demonstration that CRISPRi/a rewiring of metabolism coupled with cell free systems. This is important as it can potentially expedite the cycle of design-build-test for cell free systems, however this should be demonstrated for itaconic acid and/or glycerol production also. (See comment below)

We appreciate the positive feedback from the reviewer, their recognition of the significant impact of demonstrating CRISPRi/a rewiring with cell-free systems, and the opportunity to strengthen the manuscript with additional experimental data.

Major:

The authors describe the CRISPRi/a scheme in figure 1b for the BDO rewired strain and show inhibition of GPD1. However, supplemental figure 1c shows glycerol levels are unaffected between WT, BDO and BDO rewired. This is counter intuitive; can the authors explain this? To this end, the authors should validate that each guide is inhibiting or activating transcription at respective target loci (such as via quantitative PCR). It is possible that some guides contribute more to BDO rewired productivity than others, however this should be quantified to understand the ‘rewiring’ that has occurred.

The reviewer raises an interesting point, and we see how this needs additional explanation. In the revised manuscript, we have now more clearly explained the CRISPRi/a scheme and validated the impact of the gRNAs with quantitative PCR (qPCR).

Specifically, qPCR experiments were carried out to assess the impact of the rewiring at the transcriptional level. Our data show greater differences in threshold cycle (ΔC_t) between byproduct genes and the reference gene, ALG9 (Supp. Fig. 1b), after expression of CRISPR effectors. This indicates that all 3 ADH genes and GPD1 were downregulated by the CRISPR rewiring strategy, while BDH1 was not significantly upregulated. The unaltered glycerol titers in cell-free reactions despite reduced GPD1 expression are counterintuitive, but literature suggests several factors that could be at play. Although the cytosolic glycerol 3-phosphate dehydrogenase (GPD1) appears to be the dominant isozyme (<https://onlinelibrary.wiley.com/doi/abs/10.1111/j.1365->

2958.1995.mmi_17010095.x), knocking out the corresponding gene only slightly reduces glycerol production (<https://link.springer.com/article/10.1186/1475-2859-11-68>) which is consistent with our observed results from the dCas9-mediated transcriptional repression (Supp. Fig. 1c and Supp. Fig. 5). Expression of the mitochondrial isozyme (GPD2) could be decreased as well, but simultaneously decreasing the predominant *Adh* and *Gpd* genes leads to osmotic stress and NADH recycling deficiencies that might not be sufficiently balanced by heterologous NoxE expression and could, therefore, decrease flux toward BDO (<https://link.springer.com/article/10.1186/1475-2859-11-68> and <https://www.ncbi.nlm.nih.gov/pmc/articles/PMC3165387/>). Overflow metabolism could also be difficult to overcome through transcriptional rewiring with the large bolus of glucose added to cell-free reactions.

With this in mind, we added Supp. Fig. 1, Supp. Table 7, and the following text to the manuscript:

Supp. Fig. 1. Comparison of growth rates and gene expression. **a** Metabolic rewiring does not alter growth of BDO strain. **b** qPCR results for genes targeted by CRISPR effectors with 2-3 technical replicates from 3 biological replicates. Higher ΔCt value indicates lower expression.

Supp. Table 7. Primers for qPCR used in this study.

Target	Direction	Sequences
ADH1	Forward	TATCTTCTACGAATCCCACGG
	Reverse	CTTTGGCTTTGGAAGTGG
ADH3	Forward	GCCATTACCTGTAAACTACCA
	Reverse	TTTGACAACACTACACCAGCAC
ADH5	Forward	CGTTAAGGGCTGGAAAGTC
	Reverse	CATGCAAGTCCCATTCAACC
BDH1	Forward	CTAATCACTGGTAAGCAAAGGA
	Reverse	CCATCAACTCTTGGAAATCCC
GPD1	Forward	AAGTTCACGAATGGTTGGA
	Reverse	ACGGCTTCAAATAATGGGA
ALG9 (Reference)	Forward	GCTCCTATAGCCGTCTACGAGC
	Reverse	CTGGCAGCAGGAAAGAACTTGG

Additional manuscript text:

Page 6 – “Downregulation of transcription for byproduct genes was confirmed by qPCR, but *BDH1* was not significantly upregulated (**Supp. Fig. 1b**).”

Pages 6-7 – “Despite the confirmed decrease in *GPD1* transcription after CRISPR-mediated rewiring (**Supp. Fig. 1b**), reactions with rewired BDO extracts produced comparable glycerol titers to reactions with unmodified BDO extracts (**Supp. Fig. 2b**). We targeted *GPD1* as the dominant, cytosolic isozyme of glycerol-3-phosphate dehydrogenase, but the mitochondrial *GPD2* isozyme could also lead to glycerol production³. However, simultaneously decreasing abundance of ADH and GPD enzymes leads to osmotic stress and NADH recycling deficiencies that might not be sufficiently balanced by heterologous NoxE expression and could, therefore, decrease flux toward BDO^{4,5}. Overflow metabolism could also be difficult to overcome through transcriptional rewiring with the large bolus of glucose added to cell-free reactions. These considerations led us to use ethanol as the primary readout for byproduct formation during BDO biosynthesis.”

The authors present a framework for CRISPRi/a rewiring coupled with cell free systems however they only provide the one example of BDO production. It is possible that BDO is unique and CRISPRi/a optimization is not as effective for other pathways in cell free systems. For instance, the authors should apply a similar CRISPRi/a rewiring methodology to itaconic acid and/or glycerol production.

We agree that generalizability of the CRISPR rewiring approach is worth demonstrating here in addition to the overall cellular / cell-free framework. Per the reviewer’s suggestion, we applied this approach to glycerol production. Strains for glycerol production were rewired with CRISPR-dCas9-VPR and gRNAs targeting *GPD1* and *GPP1*, and extracts were prepared as described in the manuscript. Reactions containing extract from the rewired strain produced ~20% more glycerol than reactions with extract from the control strain without gRNAs. Although this validates the impact of *in vivo* CRISPR-dCas9 rewiring on *in vitro* biosynthesis, these extracts were unable to consume high concentrations of glucose. For this reason, the manuscript still showcases the extracts generated from strains rewired through plasmid overexpression of *GPD1* and *GPP1* to maximize cell-free glycerol titers. However, Supplementary Figure 7a and the following text were added to the manuscript to ensure the generalizability of our approach is captured.

Supp. Fig. 7. Optimization of glycerol biosynthesis. **a** Metabolic rewiring via CRISPR effectors increased glycerol production ~20%, these extracts could not consume high concentrations of glucose. Extracts from strains modified by plasmid overexpression were utilized for further glycerol biosynthesis reactions.

Page 14 – “An initial pair of glycerol strains was generated through CRISPR-mediated metabolic rewiring with guide RNAs targeting positive effectors to upregulate *GPD1* and *GPP1*², and the resulting extract from the rewired strain produced 18.7% more glycerol than extract from the control strain (**Supp. Fig. 7a**). This demonstrates generalizability of the *in vivo* CRISPR-dCas9 rewiring to increase *in vivo* biosynthesis, but these extracts were unable to consume high concentrations of glucose. For this reason, we carried out further glycerol biosynthesis reactions with extracts from the plasmid-rewired strains to maximize cell-free glycerol titers.”

Minor:

The authors should provide a supplemental table with guide RNA sequences. This is important for thorough documentation.

gRNA sequences and accessory sequences used for metabolic rewiring mediated by CRISPR-dCas9 are now provided in Supplementary Tables 5 and 6.

Supp. Table 5. Guide RNA sequences used in this study. Guide RNAs were used to regulate the expression of target targets. For repression, the guide RNA targets the ORF of genes (+) at the non-template strand. For activation, the guide RNA targets the promoter region of genes (-) at the template strand.

Targets	Regulation	Location (NGG Relative to ATG)	Sequences
For BDO Rewiring			
ADH1	Repression	+75	AATTCGTTGGCCTTTGGCTT
ADH3	Repression	+73	AGTCTTAGGGATTGCAGCTG
ADH5	Repression	+84	ATTTCGTTAGGCTTAGGTTC
GPD1	Repression	+96	CCAATCACAGTAACCTTGAA
BDH1	Activation	-207	CCTATTCTTTCCTCCTTACG
For Glycerol Rewiring			
GPD1	Activation	-178	AACCTAATTCGCACGTAGAC
GPP1	Activation	-427	ATTGCAGGATTCTCATTGTC

Supp. Table 6. Accessory sequences for sgRNA constructs. To rewire the cells for improved BDO production, a sgRNA cassette with tRNA-sgRNA bricks was used. The sequences of tRNAs and the whole cassette is as below. The expression is under the control of TEF1 promoter.

Cassettes	Sequences
RNA Scaffold	GTTTTAGAGCTAGAAATAGCAAGTTAAAATAAGGCTAGTCCGTTATCAACTT GAAAAAGTGGCACCGAGTCGGTGCTTTT
tRNA (tTCT)	GCTCGCGTGGCGTAATGGCAACGCGTCTGACTTCTAATCAGAAGAATATGG GTTTCGACCCCCATCGTGAGTG
tRNA (tCTT)	GCCTTGTTGGCGCAATCGGTAGCGCGTATGACTCTTAATCATAAAGGtTAGG GGTTCGAGCCCCCTACAGGGCT
tRNA (tGTT)	GACTCCATGGCCAAGTTGGTtAAGGCGTGCGACTGTTAATCGCAAGAATCGTG AGTTCAACCCTCACTGGGGTCG
tRNA (tGTC)	TCCGTGATAGTTTAATGGTcAGAATGGGCGCTTGTTCGCGTGCCAGaTCGGG GTTCAATTCCCCGTCGCGGAG
tRNA (tGCC)	GCGCAAGTGGTTTtAGTGGTAAAATCCAACGTTGCCATCGTTGGGCCCCCG GTTTCGATTCCGGGCTTGCGCA
tRNA (tTTC)	TCCGATATAGTGTAACGGCtATCACATCACGCTTTCACCGTGGAGaCCGGG GTTTCGACTCCCCGTATCGGAG
Whole sgRNA Cassette for BDO Rewiring	ACTAGTAGTAGCTCGCGTGGCGTAATGGCAACGCGTCTGACTTCTAATCAG AAGAATATGGGTTTCGACCCCCATCGTGAGTGCCTATTCTTTCTCCTTACGG TTTTAGAGCTAGAAATAGCAAGTTAAAATAAGGCTAGTCCGTTATCAACTTG AAAAAGTGGCACCGAGTCGGTGCTTTTGTAGTAATAGCCTTGTGGCGCA ATCGGTAGCGCGTATGACTCTTAATCATAAAGGtTAGGGGTTTCGAGCCCCCTA CAGGGCTCCAATCACAGTAACCTTGAAGTTTTAGAGCTAGAAATAGCAAGTT AAAATAAGGCTAGTCCGTTATCAACTTAAAAAGTGGCACCGAGTCGGTGC TTTTGCTAGTAAGTACTCCATGGCCAAGTTGGTtAAGGCGTGCGACTGTTA ATCGCAAGAATCGTGAGTTCAACCCTCACTGGGGTTCGATTTTCGTTAGGCTTAG GTTTCGTTTTAGAGCTAGAAATAGCAAGTTAAAATAAGGCTAGTCCGTTATCA ACTTAAAAAGTGGCACCGAGTCGGTGCTTTTGTAGTTTCTTCCGTGATA GTTTAATGGTcAGAATGGGCGCTTGTTCGCGTGCCAGaTCGGGGTTCAATTC CCCGTCGCGGAGAGTCTTAGGGATTGCAGCTGGTTTTAGAGCTAGAAATAG CAAGTTAAAATAAGGCTAGTCCGTTATCAACTTAAAAAGTGGCACCGAGTC GGTGCTTTTGTAGTATAAGCGCAAGTGGTTTtAGTGGTAAAATCCAACGTTG CCATCGTTGGGCCCCCGTTTCGATTCCGGGCTTTCGCAAAATTCGTTGGCC TTTGGCTTGTTTTtAGAGCTAGAAATAGCAAGTTAAAATAAGGCTAGTCCGTT ATCAACTTAAAAAGTGGCACCGAGTCGGTGCTTTTGTAGTATCATCCGAT ATAGTGAACGGCtATCACATCACGCTTTCACCGTGGAGaCCGGGGTTTCGA CTCCCCGTATCGGAG

References in Quoted Text

- 1 Deaner, M., Holzman, A. & Alper, H. S. Modular Ligation Extension of Guide RNA Operons (LEGO) for Multiplexed dCas9 Regulation of Metabolic Pathways in *Saccharomyces cerevisiae*. *Biotechnol J*, e1700582, doi:10.1002/biot.201700582 (2018).
- 2 Deaner, M. & Alper, H. S. Systematic testing of enzyme perturbation sensitivities via graded dCas9 modulation in *Saccharomyces cerevisiae*. *Metab Eng* **40**, 14-22, doi:10.1016/j.ymben.2017.01.012 (2017).
- 3 Eriksson, P., Andre, L., Ansell, R., Blomberg, A. & Adler, L. Cloning and characterization of GPD2, a second gene encoding sn-glycerol 3-phosphate

- dehydrogenase (NAD⁺) in *Saccharomyces cerevisiae*, and its comparison with GPD1 *Molecular Microbiology* **17**, 95-107, doi:https://doi.org/10.1111/j.1365-2958.1995.mmi_17010095.x (1995).
- 4 Ng, C. Y., Jung, M. Y., Lee, J. & Oh, M. K. Production of 2,3-butanediol in *Saccharomyces cerevisiae* by in silico aided metabolic engineering. *Microb Cell Fact* **11**, doi:<https://doi.org/10.1186/1475-2859-11-68> (2012).
- 5 Hubmann, G., Guillouet, S. & Nevoigt, E. Gpd1 and Gpd2 fine-tuning for sustainable reduction of glycerol formation in *Saccharomyces cerevisiae*. *Appl Environ Microbiol* **77**, 5857-5867, doi:10.1128/AEM.05338-11 (2011).

Reviewers' Comments:

Reviewer #1:

Remarks to the Author:

The authors have addressed the required revisions.

Reviewer #2:

Remarks to the Author:

I have no more concerns. The authors fully addressed my comments and I recommend for its publication.